# Blood Metabolic Biomarkers of Diabetes Mellitus Type 2 in Aged Adults Determined by a UPLC-MS Metabolomic Approach

**DOI:** 10.3390/metabo15060395

**Published:** 2025-06-12

**Authors:** Alba Simón, Daniel Bordonaba-Bosque, Olimpio Montero, Javier Solano-Castán, Irma Caro

**Affiliations:** 1Centro de Salud de Almudevar, Servicio Aragonés de Salud, 22270 Huesca, Spain; alba1986_7@hotmail.com; 2Servicio de Apoyo Metodológico, Estadístico y Documental (SAMEyD), Instituto Aragonés de Ciencias de la Salud (IACS), Avda. San Juan Bosco, 13, 50009 Zaragoza, Spain; dbordonaba.iacs@aragon.es; 3Unidad de Excelencia Instituto de Biomedicina y Genética Molecular de Valladolid (IBGM), Universidad de Valladolid y Consejo Superior de Investigaciones Científicas (CSIC), 47003 Valladolid, Spain; 4Máster en Nutrición Geriátrica, Departamento de Pediatría e Inmunología, Obstetricia y Ginecología, Nutrición y Bromatología, Psiquiatría e Historia de la Ciencia, Universidad de Valladolid, 47005 Valladolid, Spain; macjavi99@hotmail.com (J.S.-C.); irma.caro@uva.es (I.C.)

**Keywords:** diabetes mellitus type 2, elderly people, metabolomics, diet, UPLC-MS

## Abstract

**Background/Objectives:** Type 2 diabetes mellitus (T2DM) is a metabolic disease whose importance rises with aging, though it is also looming large in younger populations due to increasing obesity. Its effects may damage renal and heart functioning. Plasma biomarkers of T2DM have been shown through metabolomic studies under different conditions, mainly obesity, but untargeted metabolomic studies on T2DM are lacking for elderly people. **Methods:** A UPLC-MS-based metabolomic approach was conducted to ascertain potential plasma biomarkers in a cohort older than 65 years. **Results:** The dipeptide Gly-His, along with diverse lysophosphatidylcholines (LPCs), mainly LPC(14:0) and LPC(20:4), and three gangliosides were found to have different plasma content in T2DM subjects compared to control (non-diabetic) subjects (NT2DM). LPC(20:4) exhibited a gender dependence, with statistically significant differences only in females. Gly-His correlated with MEDAS-14, whereas LPC(14:0) correlated with sugar-rich food consumption. **Conclusions:** As previously demonstrated for other conditions, mainly obesity, altered lipid metabolism was shown in this study to be a hallmark of T2DM in elderly people also.

## 1. Introduction

Population aging has become a relevant concern in most Western countries. Aging is associated with diverse metabolic malfunctions that will likely lead to cardiovascular and renal diseases in the short term. Though not exclusive to the elderly, one of these age-associated diseases is type 2 diabetes mellitus (T2DM), which is a chronic metabolic disorder whose prevalence rises with age; however, its impact is also rising in younger populations owing to obesity. This disease is characterized by marked insulin resistance and strong deficient secretion, a situation that renders persistent insulin deficit and hyperglycemia [1,2]. Elderly people tend to accumulate fat in ectopic tissues, mainly liver and skeletal muscle, a factor that may elicit insulin resistance or impaired insulin secretion and ultimately T2DM. Reallocation of fat that leads to an increase in visceral adiposity is one of the multiple factors involved in the development of T2DM in elderly people, but it is accompanied by a reduction in muscular mass (sarcopenia), lower sensitivity of pancreatic beta cells to incretins, pronounced rise in insulin resistance, decreased kidney function, lower physical activity, and comorbidities that imply the use of drugs with a potential diabetogenic effect [1,3,4,5]. Thus, genetic and behavioral factors like high-carbohydrate-based nutrition together with lack of exercise predispose to T2DM development [6,7]. However, the complex molecular mechanisms involved in its development are still not well known [8]. According to data in the *Diabetes Atlas* published by the International Diabetes Federation (11th Edition) [1], the global prevalence of diabetes in adults aged between 20 and 79 years is foreseen to reach 13% of world population in 2050 (about 852.5 million people), and the percentage might be higher because of undiagnosed people [9]. The economic burden of T2DM treatments is a major concern for national health systems.

Metabolomics is an analytical methodology that allows drawing a real time image of the physiological state in a given moment [6,10]. By comparing the metabolic snapshot of T2DM patients with that of healthy individuals, insights can be gained into the metabolic pathways that become affected in the development of T2DM in unhealthy individuals and concurrently discover biomarkers that could help early diagnosis and treatment. A number of studies have used metabolomic approaches, either targeted or untargeted, to seek new biomarkers to define a more complete picture of T2DM [6,10,11,12,13,14]. Elevated branched-chain amino acids (BCAAs) and aromatic amino acids (AAAs) in blood have been observed in T2DM patients, a fact that is due to their reduced catabolism with age [11,15,16,17]. A recent study involved the cAMP response element-binding protein (CREB)-regulated transcription coactivator (CRTC) 2 in the regulation of metabolic homeostasis in visceral white adipose tissue (VAT), the role of which in the perturbation of BCAA catabolism leads to an age-associated metabolic decline [16]. Even though the link of altered lipid metabolism to insulin resistance was known a long time ago, the exact molecular mechanisms remain to be unveiled [8]. Feng et al. [14] used two lipidomic approaches, first an untargeted one, and afterwards, taking into account the results of this first approach, they devised a targeted approach to compare the lipid profile of T2DM patients with that of healthy controls. The participants in that study were males aged 35–65 years, and were classified into three groups according to their T2DM state, namely high risk, recently diagnosed, and diagnosed for more than two years. The results showed specific lipid species whose content was altered in T2DM patients in comparison to healthy controls. The lipid species they reported having altered concentrations in T2DM individuals were mainly those bearing the phosphocholine group, ceramides, short-chain saturated triglycerides, and hydroxylinoleic cholesteryl esters.

T2DM is accompanied by other comorbidities in elderly people, a situation that can complicate treatment application and response [18]. In this regard, metabolomics may help to disclose differences in altered metabolic pathways in younger people and guide personal treatment for elderly patients [12,16,19,20,21]. In this prospective study, the metabolic profile of T2DM patients aged more than 65 years was compared with that of non-diabetic individuals within a similar age range in an untargeted metabolomic study.

## 2. Materials and Methods

### 2.1. Chemicals and Reagents

Methanol and acetonitrile were OPTIMA^®^ LC-MS grade from Merck (Darmstadt, Germany). Formic acid was pro analysi ACS, Reag. Ph Eur from Merck (Darmstadt, Germany). Ammonium acetate was pro analysi ACS, Reag. Ph Eur from Merck. Ammonia solution 32% was from Merck (Darmstadt, Germany). Water was lab-produced with Milli-Q equipment (Darmstadt, Germany).

### 2.2. Population Characteristics and Recruitment

Fifty-nine subjects aged ≥ 65 years (mean age 73.6 ± 8.8 years) were recruited at a medical center in Calaceite (Teruel, Spain). The volunteers comprised a non-diabetic control group (NT2DM) of 21 females and 11 males with a mean age of 71.6 ± 9.1 years and 15 females and 12 males diagnosed with diabetes (T2DM) with a mean age of 75.9 ± 8.1 years and average disease duration of 5 years. The medication included oral antidiabetic drugs like metformin, alone or in combination with vildagliptin, sitagliptin, or dapagliflozin, or repaglinide or gliclazide, in addition to insulin. The clinical and other characteristics of the cohort are depicted in Table 1. Information on ethical concerns is stated below.

### 2.3. Metabolomic Experiments

#### 2.3.1. Sample Harvesting and Processing

Blood samples from all participants were obtained by puncture of the forearm and collected in tubes containing lithium heparin. Plasma was harvested after centrifugation of tubes containing the blood at 738 g and 4 °C for 10 min, and 500 μL was transferred to Eppendorf containers and kept at −80 °C until UPLC-MS measurements.

For metabolite extraction, 100 μL of plasma was mixed with 500 μL of cold acetonitrile, vortexed, and kept at 4 °C for 1 h. Afterwards, the mixture was centrifuged at 1660× *g* and 4 °C for 10 min, and 500 μL of supernatant was transferred to new containers. Three blanks for the UPLC-MS measurements were prepared using 100 μL of Milli-Q water instead of plasma. Additionally, quality control (QC) samples (in triplicate) were prepared by pooling 50 μL each of plasma samples, of which 100 μL was processed in the same way as the individual plasma samples.

#### 2.3.2. UPLC-MS Measurements

These measurements were conducted as in Albillos et al. [22]. An Acquity UPLC HSS T3 1.8 µm, 2.1 × 100 mm column was used for the liquid chromatographic separation of the extracted compounds. The chromatography was carried out in an Acquity Ultraperformance LC (UPLC) from WATERS (Barcelona, Spain) with an eluent flow of 0.35 mL/min, and 7.5 µL of each sample was injected. In order to disperse error propagation, plasma samples were randomly injected. The blanks were injected at the onset of the chromatographic analysis, and quality control samples were injected next as well as after every 20 sample injections. Two solvents were used for the elution gradient, with solvent A being methanol–water (2:8, *v*/*v*) + 0.1% formic acid, and solvent B being 100% acetonitrile + 0.1% formic acid, with steps of initial, 99.9% A; 1 min, isocratic; 3.5 min, 20% A; 5 min, isocratic; 9.5 min, 0.1% A; 11.0 min, isocratic; 14.0 min, 99.9% A; and 15.0 min, isocratic. The column eluent was directly connected to a mass spectrometer (SYNAPT HDMS G2, WATERS, Barcelona, Spain) fitted with an electrospray ionization source (ESI, Z-spray^®^) and time-of-flight analyzer (ESI-QToF-MS). An MS^E^ method was used for the analysis, in which data were acquired within the *m*/*z* range of 100 to 1200 Da under the low-energy function (function 1), it being full-scan equivalent, and within the *m*/*z* range of 50 to 1000 Da under the high-energy function (function 2) with non-specific fragmentation of base peak *m*/*z* values detected in the full scan. Data were acquired in resolution mode (expected error of less than 3 ppm corresponding to a minimal resolution of 20,000) using MassLynx^®^ software version 4.1 (SCN871 package, WATERS, Manchester, UK). The QToF-MS was calibrated using 0.5 mM sodium formate in 9:1 (*v*/*v*) 2-propanol–water, and as a reference, 2 ng/µL leucine-enkephalin (Leu-Enk) in 50:50 (*v*/*v*) acetonitrile–water with 0.1% formic acid was used. The metabolites were detected with positive ionization, and the parameters were capillarity 2.5 KV, cone 25 V, source temperature 100 °C, desolvation temperature 320 °C, cone gas (nitrogen) 35 L/h, and desolvation gas flow (nitrogen) 700 L/h.

#### 2.3.3. UPLC-MS Data Analysis

Using MarkerLynx^®^ software V4.1 (SCN871 package, WATERS, Manchester, UK) the UPLC-MS data were processed and the variables plasma sample (including blanks), retention time *m*/*z* values (molecular features), and normalized (scaled to Pareto variance) signal intensity of the *m*/*z* value were arranged in a three-dimensional Pareto-scaled data array. Following this, *m*/*z* values considered noise or contaminants were manually excluded. Original data are included in an Excel file and reported in Appendix A.

The Extended Statistics^®^ (XS) application included in MarkerLynx^®^ V4.1 (SCN871 package) and the freely online-accessible MetaboAnalyst 6.0 (https://www.metaboanalyst.ca/ accessed on 12 July 2024) were used to conduct multivariate statistical analysis on the generated data array. Normalization by sum and Pareto scaling was chosen in both programs. Binary comparisons were conducted for the two sample groups: T2DM of diagnosed patients and NT2DM (undiagnosed control group). Cross-validation (10-fold) and 1000 permutations were run for each comparison after removal of outliers. The QuantLynx^®^ application (SCN871 package, Waters, Manchester, UK) was used to quantify the chromatographic peak area of the selected metabolites as potential biomarkers. Values of *m*/*z* with a defined chromatographic peak were only accepted to potentially arise from any true compound. Selected differential metabolites were tentatively identified by comparison of the elemental composition measured in the mass spectrum with that available in the free accessible databases HMDB, METLIN, KEGG, ChEBI, and LipidMaps. Additionally, comparison of relative retention time and adduct formation for those compounds that had been identified in previous studies (lysophosphatidyl cholines for instance) was taken into consideration. Specific fragments in the MS^E^ spectrum, either known (*m*/*z* 184.07 and the fragment corresponding to the neutral loss of the acyl group for phosphatidyl cholines and sphingomyelins, for instance) or visualized in the HMDB for the given compound, were used to validate the compound identification. For gangliosides, the nominal mass (M) calculated from the *m*/*z* value of the spectrum first peak and taking into consideration that ionization was +2 (i.e., [M + X + Y]^2+^, where X and Y may be equal or different and account for H, NH_4_^+^ or both), was used to find whether it corresponded to any compound in the LipidMaps database, but a specific species could not be determined and only some are tentatively suggested and indicated where appropriate.

#### 2.3.4. Classical Statistical Analysis

A descriptive analysis of the data was performed. Qualitative variables are presented using frequency distributions and percentages for each category. For quantitative variables, measures of central tendency (mean or median), and dispersion (standard deviation or percentiles) were calculated.

The association between variables was evaluated using hypothesis testing. For qualitative variables, proportions were compared using the chi-squared test, or when necessary, Fisher’s exact test. For quantitative variables, the Shapiro–Wilk normality test was applied to determine whether they followed a normal distribution. Since they did not meet this assumption, nonparametric tests were used for group comparisons.

To analyze the relationship between variables while controlling for potential confounding factors, multivariate regression models were used. Some variables, such as MEDAS, were categorized to facilitate the interpretation of coefficients and to reduce bias in parameter estimation due to low-frequency categories. Odds ratios with 95% confidence intervals were calculated for each variable.

Statistical significance was regarded as *p* < 0.05, and all tests were two-tailed.

The statistical analysis was performed using R software version 4.4.0 (R Core Team, 2024), together with the packages *dplyr* [23] and *gtsummary* [24] for data manipulation and generation of descriptive tables. Appropriate tests were applied according to variable type and distribution.

## 3. Results

### 3.1. Evaluation of Cohort Characteristics

The cohort characteristics are shown in Table 1. Age was significantly higher in the T2DM group (*p* > 0.004). Females were predominant in both groups, and in particular in the NT2DM group, where they accounted for 66%. There were no significant differences in body mass index (BMI) between the groups, but the number of individuals with obesity was higher in the control (NT2DM) group than in the T2DM group (14 vs. 9).

Regarding lifestyle, a high proportion of T2DM group individuals (78%) did not consume alcohol, and non-smokers were predominant in both groups. Both groups reported comparable distribution regarding physical activity, this being measured as MET h/week, with vigorous-intensity activity being the most common in both groups. Adherence to the Mediterranean diet (MEDAS-14) was also comparable in the groups, with no significantly different scores. Sugar intake was significantly lower in the T2DM group (*p* = 0.013), with only 33% reporting high sugar intake (>36 g/day for men and >25 g/day for women). Only five NT2DM and six T2DM subjects ate a meat-free diet, with no significant differences between groups regarding high fat consumption and meat intake patterns.

Familial endocrine disease was more common in the T2DM group than in the NT2DM group (70% vs. 25%; *p* < 0.001), whereas a family history of cardiovascular disease was more prevalent in the control group (47% vs. 26%). Twenty-four T2DM individuals (89%) against twelve NT2DM individuals (38%) had an intake of three or more drugs, but only twelve of the T2DM patients (44%) and eleven of the controls (34%) had prescriptions for dyslipidemia medication.

Concerning biochemical parameters, only fasting glucose, total serum cholesterol, and LDL cholesterol showed statistically significant differences, the cholesterol-related parameters being higher in the NT2DM group, while the opposite was found for fasting glucose, likely because 44% of the T2DM group were receiving lipid-lowering therapy.

### 3.2. UPLC-MS Data Analysis

As indicated previously, participants were grouped as T2DM for type 2 diabetes mellitus-diagnosed patients and NT2DM for T2DM-free individuals (controls). Typical chromatograms for both groups are shown in Figure 1. Subtle variations in peak intensities can be observed between the groups in the chromatograms.

Results were comparable in the XS application and MetaboAnalyst 6.0. Principal component analysis (PCA) did not render good separation of the two groups. In contrast, both groups were clearly separated when partial least squares discriminant analysis (PLS-DA) was used for the multivariate statistical analysis (Figure 2). Correlation coefficients obtained in the XS application were R2Y (cum) = 0.80 and Q2 (cum) = 0.71 for component 1, and R2Y (cum) = 0.91 and Q2 (cum) = 0.87 for component 2. According to MetaboAnalyst output, components 1 and 2 explained 28.9% of variance. This is not high, but enough to achieve full separation and good prediction capacity of the model according to Q2 and R2 parameters.

Fourteen metabolites were pointed out as potential biomarkers according to the PLS-DA in both the XS application and MetaboAnalyst (VIP score > 1.0): seven lysophosphatidylcholines (LPCs), one lysophosphoinositide (LPI), three gangliosides, two phosphatidylcholines (PCs), and one dipeptide (Gly-His). These were tentatively identified taking into consideration the *m*/*z*, elemental composition, and retention time. Additionally, compounds could not be ascribed to four features that were also shown as differential ones in the PLS-DA and remained unidentified. The ganglioside species provided in the footnote of Table 2 were obtained from the LipidMaps database, but could not be verified. Apart from LPC(20:4) all metabolites exhibited lower values in the T2DM group than in the NT2DM group (negative value of Log_2_ FC). However, only LPCs 14:0, 16:0, 18:0, and 18:2, gangliosides 1 and 2, Gly-His, and the unidentified compound showed significant differences between the groups according to the chromatographic peak area normalized to that of the reference compound reserpine (unpaired *t*-test).

Classical statistical analysis was conducted to find out whether the potential biomarkers arising from the PLS-DA showed any correlation between them and with the cohort characteristics reported in Table 1. Positive correlations were shown between the lipids (LPCs and the ganglioside 2), whereas negative correlations were obtained for the dipeptide Gly-His with the lipids apart from LPC(14:0) (see Table 3). Significant correlations (*p* < 0.05) were found for Gly-His with LPC(22:6) and LPC(14:0). Ganglioside 2 correlated significantly with the three LPCs, and LPC(22:6) significantly correlated with LPC(20:4).

Significant correlations of the selected biomarkers with some of the cohort characteristics are shown in Table 3 (*p* values determined after Wilcoxon rank-sum exact test). Gly-His correlated with MEDAS-14 (*p* = 0.004), with increasing values from strong to low, and LPC(14:0) correlated positively with consumption of sugar-rich foods (*p* = 0.012) (see Appendix A). Though not significant (*p* = 0.075), consumption of sugar-rich foods also rendered a higher value of ganglioside 2. LPC(20:4) was significantly (*p* = 0.005) higher in males (1718.3 ± 741.2) than in females (1217.4 ± 604.7).

Ganglioside 2, which showed higher values in the NT2DM group than in the T2DM group, exhibited significant differences between groups only for males (*p* = 0.02), with mean values of the medians of 82.69 and 55.02 in the NT2DM and T2DM groups, respectively. A somewhat similar result was found for LPC(14:0), with mean values of the medians of 253.26 and 126.34 in the NT2DM and T2DM groups for males. Results from univariate and multivariate analyses showed that the gender variable influenced the statistical significance of differences between groups (Table 4), at least to some extent. In particular, significant differences between groups of LPC(20:4) were clearly bound to the other variables and specifically to gender. Thus, a p value of 0.100 resulted in the univariate analysis, whereas p values of 0.049 and 0.026 were obtained in the multivariate analysis with and without gender, respectively. Nonetheless, the odds ratio (OR) was always slightly higher than 1 (1.0007 in the univariate analysis and 1.0016 or 1.0018 in the multivariate analysis).

Hierarchical cluster analysis was also conducted for the selected metabolites illustrated in Table 2 (Figure 3). The general cluster was split into two major sub-clusters, which included small clusters containing samples from both T2DM and NT2DM2 groups. This partial mixing may be a reflection of the data dispersion found for some metabolites, a fact that led to partial intersection of the groups. The partial intersection is also reflected in the heatmap, but enhanced metabolite content was found in the NT2DM group.

## 4. Discussion

Amino acids have been shown to play an essential role in the pathogenesis of diabetes [11]. Indeed, high plasma or serum levels of branched-chain (BCAAs) and aromatic (AAAs) amino acids have been found in different studies to be reliable indicators of insulin resistance and pre-diabetes state [6,17,25]. In contrast, high serum levels of glutamine (Gln) and histidine (His) were proposed to be indicative of lower risk of incident diabetes mellitus type 2 (T2DM) in obese subjects [20], and oral supplementation of histidine was shown to improve IR [26,27]. In a study by Gu et al. [20], dipeptides containing glycine (Gly) showed reduced levels in diabetic obese patients compared with non-diabetic obese individuals. The results of our study show that a dipeptide, namely Gly-His, whose plasma content was lower in diabetic patients than in the controls, might be a biomarker for high risk of T2DM development when its level drops below a given value. Indeed, reduced Gly content is considered a biomarker for insulin resistance and subsequent T2DM development [28,29,30,31,32]. Glycine takes part in a number of pathways that may decrease its availability, especially gluconeogenesis and glutathione (GSH) synthesis [33], a fact that leads to chronic deficiency in GSH in elderly people, and indeed it was shown that a diet supplemented with Gly (and cysteine, Cys) could raise the GSH level in 82-week-old mice up to that measured in 22-week-old mice, as well as in old humans compared to young humans [19], respectively. GSH deficiency may cause malfunction of mitochondria due to excessive oxidative stress and increased lipid oxidation.

A number of studies have shown that lipid metabolism is altered in subjects with T2DM pathology [6,7,14,34], and even long before disease manifestations [35]. In agreement with other published studies, our results show that some lysophosphatidylcholines (LPCs) have reduced levels in T2DM patients compared with non-diabetic subjects. In particular, we found that the circulating myristoyl-glycerophosphocholine (1-tetradecanoyl-*sn*-glycero-3-phosphocholine, LPC(14:0) was significantly reduced in T2DM patients (Table 2). This metabolite had previously been proposed by Ha et al. [34] as a relevant biomarker, even showing a positive correlation with arterial stiffness; however, contrary to the finding of Ha et al., we found its level was lower in the T2DM group than in the NMD2 group. Similar results were found in this study for other saturated LPCs (16:0 and 18:0), as well as for LPC(18:2). Ferrannini et al. [30] and Wang-Sattler et al. [36] also pointed out decreased levels of this latter LPC (linoleoyl-glycerophosphocholine, L-GPC) in patients as indicative of IR and T2DM. Conversely, Feng et al. [14] did not report LPC(18:2) as a potential biomarker, but did report increased levels in diabetic subjects of the hydroxy-linoleic acid cholesteryl ester (CE(18:2-OH)), a fact that the authors attribute to enhanced oxidative stress. Reduced PC(16:0/18:2) in the T2DM group was also found in this study, as well as in that by Feng et al. for the high-risk group compared to controls, which could be indicative of the relevance of linoleic acid in diabetes mellitus. Indeed, hydroxy-linoleic acid acts as an agonist of peroxisome proliferator-activated receptor γ (PPARγ), which is implicated in inflammation, IR, and glucose metabolism [37].

LPC(20:4) exhibited an opposite pattern to that of the aforementioned LPCs, with values higher in T2DM individuals than in NMD2 individuals (Table 2), but only LPC(20:4) showed significant differences (*p* = 0.026 without gender and *p* = 0.049 when gender was included) in the multivariate analysis (Table 4). To our knowledge, LPC(20:4) has not previously been reported regarding IR and diabetes type 2 biomarkers. This is surprising, because arachidonic acid is positively related to inflammation through its oxidized metabolites, the eicosanoids, inflammation being associated with T2DM progress [38,39,40,41]. Nonetheless, Zhang et al. [42] found that free arachidonic acid could inhibit inflammatory responses through modulating the activity of Toll-like receptor 4 (TLR4), and it may be that the increased arachidonic acid in the form of LPC(20:4) in T2DM patients prevented the inhibitory regulation of inflammation. LPC(22:6) was evaluated as a less predictive metabolite by Ha et al. [34]. Even though the majority of LPCs are formed from oxidized phospholipids (PPLs), plasma LPC(20:4) and LPC(22:6) come from the action of human lecithin cholesterol acyltransferase (LCAT) [43]; however, low LCAT activity in diabetic patients, particularly in women, was reported [44], which is contradictory to the apparently higher levels of LPC(20:4) in the T2DM group found in this study.

Nowadays, dysregulation of lipid metabolism is accepted as one of the diabetes mellitus type 2 hallmarks, even playing a central role in its pathogenesis [14]. The significantly positive correlation of relevant LPC(20:4) and LPC(22:6) between them and with ganglioside 2 (Table 3) indicates an interplay between specific lipid pathways. PCs have a saturated fatty acyl chain esterified at the *sn*-1 position of the glycerol backbone and an unsaturated fatty acyl chain at the *sn*-2 position, and enhanced levels of these LPCs and ganglioside(s) may represent a rise in the release of saturated acyl chains for ceramide synthesis [45], which in turn would result in excess circulating glucose for ganglioside synthesis [46]. Enhanced derivation of ceramides towards gangliosides may be bound to a counteracting mechanism that tends to reduce the ceramide content. Aberrant accumulation of ceramides may block the translocation of the glucose transporter 4 through the inhibition of Akt/PKB activation, which leads to inhibition of glucose uptake and glycogen synthesis in adipocytes and isolated skeletal muscle [45,47].

Gly-His correlated positively with MEDAS-14, while LPC(14:0) did with sugar-rich foods. LPC(20:4) seems to have a gender-dependent effect (Table 4). Accordingly, it seems that different metabolites might be representative of different affected pathways or even organs, which leads to varying potential biomarkers relying on the particular cohort characteristics like age, gender predominance, and of course lifestyle. Thus, even though there seems to be a general pattern of altered lipid and amino acid metabolisms, a number of factors may output different specific biomarkers, and to unveil which biomarkers are representative of disease development and which of them correspond to evolution under treatment and at every age is necessary to actually understand the disease.

Even though data dispersion, a feature common to plasma analysis, may have led to partial mixing of samples in the hierarchical cluster analysis (Figure 3), the majority of small sub-clusters showed grouping of samples within the NT2DM or T2DM groups, with apparent enhanced metabolite levels in the NT2DM group. It should be taken into consideration that not all the metabolites showed statistically significant differences.

There are three limitations associated with this study: (i) the cohort was rather small; (ii) single-center recruitment, though from different small villages; and (iii) extensive pharmacology affecting both controls and patients. These limitations may have affected the biomarker search, but it is evident that a number of biomarkers found in this study are coincident with those found in other groups’ studies, in particular those related to lysophospholipids. Additionally, a new class of lipids is shown in our study to afford potential biomarkers, which are the gangliosides. In spite of the burden imposed by the extensive pharmacology and the small cohort, excellent group separation in the PLS-DA using two different software packages may suggest, in our opinion, robustness in our results.

## 5. Conclusions

In this study, we determined a set of biomarkers related to type 2 diabetes mellitus (T2DM) in aged people (>65 years) using an untargeted metabolomic approach. Lysophosphatidylcholines (LPCs) and gangliosides were found to be potential biomarkers, thus confirming altered lipid metabolism. The dipeptide Gly-His was also revealed as a potential biomarker, which seems to be in agreement with the reported role of amino acids in T2DM through deficient anti-oxidative pathways related to GSH and mitochondria malfunction. Some biomarkers could be specifically related to gender, like LPC(20:4), and others to diet like LPC(14:0) and ganglioside 2. Nonetheless, no correlation was shown for the biomarkers with particular diet characteristics like fat or sugar intake. Further studies addressing the roles played by age and obesity in addition to diet are required for a more complete understanding of this metabolic disease.

The results of this study open the possibility of discriminating specific biomarkers for elderly people, given that their metabolism intrinsically differs from that of the younger people, and additionally, there might be potential confounding factors bound to polypharmacy.

## Figures and Tables

**Figure 1 metabolites-15-00395-f001:**
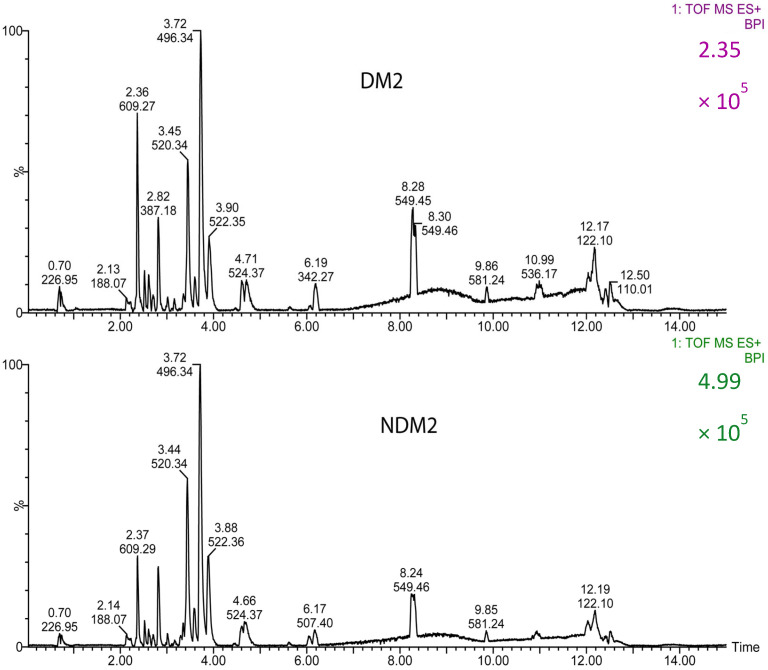
Typical base peak chromatograms (BPI) obtained for T2DM (diabetic participants) and NT2DM (non-diabetic controls) groups with positive ionization.

**Figure 2 metabolites-15-00395-f002:**
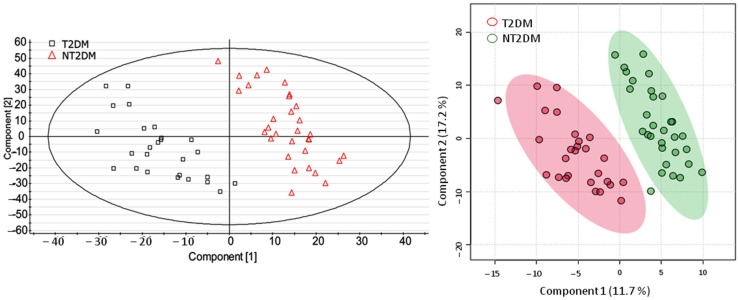
Score plots of the UPLC-MS data obtained after processing with partial least square discriminant analysis (PLS-DA) using the XS application (left panel) and MetaboAnalyst (right panel). Sample 47 was considered an outlier and excluded.

**Figure 3 metabolites-15-00395-f003:**
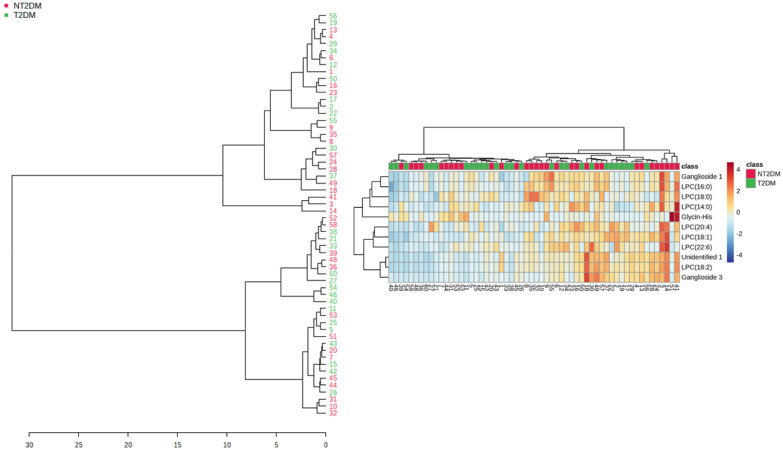
Clustering tree for the samples according to metabolite content (left panel). The corresponding heatmap is displayed in the right panel.

**Table 1 metabolites-15-00395-t001:** Characteristics of participants. NMD2 is the control. T2DM is diagnosed diabetics. Continuous variables are expressed as means ± standard deviation.

Variable	NT2DM Group (n = 32)	T2DM Group (n = 27)
Demographics and anthropometric characteristics		
Age (years; *p* = 0.004)	71.4 ± 9.0	75.9 ± 8.1
Sex (female/male)	21 (66%)/11 (34%)	15 (56%)/12 (44%)
BMI (kg/m^2^) ^1^	28.8 ± 5.8	28.5 ± 6.3
Underweight	4 (13%)	3 (11%)
Normal	7 (22%)	9 (33%)
Overweight	7 (22%)	6 (22%)
Obese	14 (44%)	9 (33%)
Waist circumference—female (cm) ^2^	100.6 ± 16.0 (n = 21)	100.6 ± 14.9 (n = 15)
Waist circumference—male (cm) ^2^	104.8 ± 10.4 (n = 11)	108.9 ± 11.4 (n = 12)
Lifestyle and dietary habits		
Alcohol intake ^3^ (*p* = 0.049)	15 (47%)	6 (22%)
Smoking status		
Never smoker	22 (69%)	20 (74%)
Former smoker	6 (19%)	5 (19%)
Current smoker	4 (13%)	2 (7.4%)
Physical activity (MET h/week) ^4^	77.1 ± 93.8	64.9 ± 67.0
Vigorous intensity	16 (50%)	14 (52%)
Moderate	10 (31%)	7 (26%)
Light	2 (6.3%)	2 (7.4%)
Rest	4 (13%)	4 (15%)
Mediterranean diet score (MEDAS-14) ^5^	8.6 ± 1.5	7.9 ± 1.1
Low adherence	7 (22%)	8 (30%)
Moderate adherence	19 (59%)	18 (67%)
Strong adherence	6 (19%)	1 (3.7%)
High sugar intake ^6^ (*p* = 0.013)	21 (66%)	9 (33%)
Sugar intake (g/day: *p* = 0.019)	47.3 ± 83.7	26.7 ± 55.5
High fat intake ^7^	13 (41%)	8 (30%)
Meat intake type		
No-meat diet	5 (16%)	6 (22%)
White and processed meat	8 (25%)	7 (26%)
Red and processed meat	8 (25%)	11 (41%)
White meat	11 (34%)	3 (11%)
Family history, treatments, polypharmacy, blood pressure, and biochemical parameters		
Family history of cardiovascular disease	15 (47%)	7 (26%)
Family history of endocrine disease (*p* < 0.001)	8 (25%)	19 (70%)
Treatment for dyslipidemia	11 (34%)	12 (44%)
Polypharmacy (≥3 medications, *p* < 0.001)	12 (38%)	24 (89%)
Systolic blood pressure, left arm (mmHg) (*p* = 0.038)	134.6 ± 19.4	138.4 ± 15.2
Diastolic blood pressure, right arm (mmHg)	83.1 ± 9.0	79.1 ± 8.5
HbA1c (%)	-	6.9 ± 0.9
Fasting glucose (mmol/L, *p* < 0.001)	5.1 ± 0.6	7.2 ± 1.9
Total serum cholesterol (mg/dL, *p* < 0.001)	193.0 ± 28.8	164.2 ± 36.3
HDL (mg/dL)	61.8 ± 16.2	61.6 ± 32.9
LDL (mg/dL, *p* < 0.001)	109.0 ± 24.2	86.4 ± 29.5
Triglycerides (mg/dL)	102.4 ± 37.3	105.4 ± 43.0
TG/HDL ratio	1.8 ± 1.0	2.0 ± 1.1
LDL/HDL cholesterol ratio	1.7 ± 0.6	1.6 ± 0.7
High cardiovascular risk	13 (41%)	11 (41%)

Values: ^1^ BMI (kg/m^2^): based on SEGG and SENPE guidelines—underweight (<18.5), normal weight (22–26.9), overweight (27–29.9), and obesity (>30); ^2^ waist circumference (cm): WHO thresholds—88 cm (women) and 102 cm (men) as the upper limit for metabolic health; ^3^ alcohol intake—considered “yes” if exceeding high-risk levels (>2 UBEs/day for women, >4 UBEs/day for men); ^4^ physical activity (MET h/week)—rest (≤4.2), light (~10–11.2), moderate (~22.4–39.2), and vigorous (>40); ^5^ Mediterranean diet score (MEDAS-14)—high adherence (≥9), moderate adherence (5–8), low adherence (≤4); ^6^ high sugar intake—>36 g/day for men, >25 g/day for women; ^7^ high fat intake—categorized as high (≥2 portion) and low (0–1).

**Table 2 metabolites-15-00395-t002:** Relevant differential metabolites shown by the PLS-DA. Chromatographic peak areas of the compounds were normalized to the chromatographic peak area of reserpine, and are indicated as means ± standard deviation. FC (Log_2_) is the logarithm in base 2 and was calculated as the ratio of the mean value of the normalized chromatographic peak area of the T2DM group to that of the NT2DM group. *p*-values were determined after statistical comparisons were performed using independent *t*-tests and the Mann–Whitney U test, as appropriate. Gangliosides and unidentified 1 have ionization +2 ([M + 2H]^2+^, [M + H + NH4]^2+^, or [M + 2NH4]^2+^). A formula for unidentified 1 cannot be proposed.

Metabolite	Formula [M + H]^+^	*m*/*z*	Normalized Chromatographic Peak Areas		Retention Time (min)	FC (Log_2_)	Regulation	*p*
			T2DM	NT2DM				
LPC(14:0)	C_22_H_47_NO_7_P	468.3072	0.032 ± 0.016	0.053 ± 0.029	3.18	−0.73	Down	<0.001
LPC(16:0)	C_24_H_50_NO_7_P	496.3413	3.321 ± 0.982	4.075 ± 0.984	3.72	−0.29	Down	0.003
LPC(18:0)	C_26_H_54_NO_7_P	525.3698	0.235 ± 0.067	0.332 ± 0.109	4.68	−0.50	Down	<0.001
LPC(18:1)	C_26_H_52_N_89_P	522.3556	1.253 ± 0.502	1.306 ± 0.494	3.90	−0.06	Down	0.344
LPC(18:2)	C_26_H_50_NO_7_P	520.3401	1.834 ± 0.794	2.320 ± 1.027	3.45	−0.34	Down	0.023
LPC(20:4)	C_28_H_50_NO_7_P	544.3397	0.323 ± 0.139	0.313 ± 0.159	3.42	+0.05	Up	0.400
LPC(22:6)	C_30_H_50_NO_7_P	569.3391	0.077 ± 0.038	0.083 ± 0.040	3.36	−0.10	Down	0.296
PC(16:0/18:2)	C_42_H_80_NO_8_P	758.5605	2.29 × 10^−4^ ± 6.55 × 10^−4^	6.22 × 10^−4^ ± 12.6 × 10^−4^	7.54	−1.44	Down	0.081
Ganglioside 1	C_75_H_137_N_3_O_27_	754.9894	0.074 ± 0.041	0.096 ± 0.045	3.72	−0.37	Down	0.032
Ganglioside 2	C_75_H_135_N_3_O_27_	762.9800	0.013 ± 0.010	0.019 ± 0.009	3.72	−0.57	Down	0.009
Ganglioside 3	C_78_H_142_N_2_O_31_	791.4910	0.017 ± 0.014	0.024 ± 0.022	3.44	−0.52	Down	0.080
Glycine-Histidine	C_8_H_12_N_4_O_3_	195.0888	0.008 ± 0.013	0.021 ± 0.035	2.18	−1.43	Down	0.040
Unidentified 1		531.3243	0.100 ± 0.056	0.133 ± 0.064	3.44	−0.41	Down	0.020

**Table 3 metabolites-15-00395-t003:** Matrix of correlations between the most relevant PLS-DA biomarkers. Correlations with MEDAS-14 are also provided. Correlations are indicated by r and *p* (in parentheses) values.

	Gly-His	LPC(22:6)	LPC(20:4)	LPC(14:0)	Ganglioside 2	MEDAS-14
Gly-His	-	−0.211 (0.113)	−0.057 (0.669)	0.366 (0.005)	0.006 (0.963)	−0.412 (0.001)
LPC(22:6)	−0.211 (0.113)	-	0.513 (<0.001)	0.250 (0.059)	0.540 (<0.001)	0.197 (0.137)
LPC(20:4)	−0.057 (0.669)	0.513 (<0.001)	-	0.190 (0.153)	0.333 (0.011)	−0.042 (0.752)
LPC(14:0)	0.366 (0.005)	0.250 (0.059)	0.190 (0.153)	-	0.480 (<0.001)	−0.037 (0.784)
Ganglioside 2	0.006 (0.963)	0.540 (<0.001)	0.333 (0.011)	0.480 (<0.001)	-	0.106 (0.429)
MEDAS-14	−0.412 (0.001)	0.197 (0.137)	−0.042 (0.752)	−0.037 (0.784)	0.106 (0.429)	-

**Table 4 metabolites-15-00395-t004:** Results of the statistical univariate and multivariate analyses for a comparison of metabolite values between T2DM and NT2DM groups. OR: odds ratio. CI: 95% confidence interval. *p*-values < 0.05 were considered statistically significant.

			Univariate Analysis			Multivariate Analysis		
Characteristics		n	OR	95% CI	*p*-Value	OR	95% CI	*p*-Value
Gender masculine		59	1.527	0.533–4.441	0.430	1.413	0.281–7.305	0.670
Gly-Hist	Gender	59	0.994	0.985–1.000	0.108	0.995	0.985–1.003	0.349
	No gender	59	0.994	0.986–1.000	0.108	0.996	0.986–1.003	0.336
LPC(22:6)	Gender	59	1.000	0.997–1.003	0.624	1.001	0.995–1.008	0.652
	No gender	59	1.001	0.998–1.004	0.624	1.001	0.995–1.008	0.740
LPC(20:4)	Gender	59	1.000	0.999–1.001	0.100	1.001	1.000–1.003	0.049
	No gender	59	1.001	0.999–1.002	0.100	1.002	1.001–1.004	0.026
LPC(14:0)	Gender	59	0.990	0.982–0.996	0.009	0.988	0.977–0.997	0.018
	No gender	59	0.991	0.983–0.997	0.009	0.989	0.978–0.997	0.019
Ganglioside 2	Gender	59	0.986	0.971–0.998	0.044	0.976	0.950–0.996	0.042
	No gender	59	0.986	0.971–0.999	0.044	0.977	0.952–0.997	0.045

## Data Availability

Original UPLC-MS data are provided as Appendix A in Excel file format.

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
