# Peer review of "Blood Metabolic Biomarkers of Diabetes Mellitus Type 2 in Aged Adults Determined by a UPLC-MS Metabolomic Approach"

_metabolites, 2025, doi:10.3390/metabo15060395_

Round 1

Reviewer 1 Report

Comments and Suggestions for Authors

Many thanks for the opportunity to analyze the manuscript from Simon and coauthors. The presented study goal is interesting and has potential scientific and clinical impact. I have a couple questions which can allow to enhance the presented manuscript.

1.Which is the study limitations? Respective subsection should be added at the end of the manuscript. For example, single center study design can be one of the limitations.

2.Why patients from the study have so low obesity level? BMI values are just a little higher than normal.

3.Why authors did not evaluate HbA1c level in normal group?

4.What is the crucial impact of untargeted metabolomics approach for further routine diagnostics? Metabolomic approach is not so cheap, may be analysis of markers concentration based on classic methods would be better?

Author Response

For research article metabolites-3647196

Response to Reviewer 1 Comments

1. Summary

2. Questions for General Evaluation

Reviewer’s Evaluation

Response and Revisions

Yes

Can be improved

Must be improved

Not applicable

Does the introduction provide sufficient background and include all relevant references?

(x)

( )

( )

( )

Is the research design appropriate?

( )

(x)

( )

( )

Are the methods adequately described?

( )

(x)

( )

( )

Are the results clearly presented?

( )

(x)

( )

( )

Are the conclusions supported by the results?

(x)

( )

( )

( )

Are all figures and tables clear and well-presented?

( )

( )

( )

( )

Many thanks for the opportunity to analyze the manuscript from Simon and coauthors. The presented study goal is interesting and has potential scientific and clinical impact. I have a couple questions which can allow to enhance the presented manuscript.

3. Point-by-point response to Comments and Suggestions for Authors

Comments 1: Which is the study limitations? Respective subsection should be added at the end of the manuscript. For example, single center study design can be one of the limitations.

Response 1: We thank the reviewer for this comment. Certainly, this issue needs to be explained more extensively. Therefore, the following paragraph is now added at the end of the discussion section (lines 429-438): “There are mostly three limitations in our study performance, which are: i) the cohort is rather reduced; ii) a single center recruitment though from different small villages; and iii) extensive pharmacology that is affecting both controls and patients. These limitations may have affected the biomarker search, but it is evident that a number of biomarkers found in this study are coincident with those found in other group’s studies, in particular those related to lysophospholipids. Additionally, a new class of lipids is shown in our study to afford potential biomarkers, which are the gangliosides. In spite of the burden imposed by the extensive pharmacology together with a reduced cohort, excellent group separation in the PLS-DA using two different softwares, which may account in our opinion for high robustness of our results.”

Comments 2: Why patients from the study have so low obesity level? BMI values are just a little higher than normal.

Response 2: All participants in this study, both patients and controls, come from a rural environment where two factors contribute to insure tight control of the disease evolution in the diabetic patients. These factors are: i) the people inhabiting the countryside is used to have an active lifestyle, with daily walks of more than 1 hour and extremely reduced driving for their displacements; and 2) the rather low ratio of patients to healthcare workers in the rural area where the people was recruited allows to keep an efficient tracking of disease evolution besides intensive promotion of a healthy lifestyle bound to activity and Mediterranean diet adhesion.

Comment 3: Why authors did not evaluate HbA1c level in normal group?

Response 3: After patient signed consent, Ms Alba Simón, who is the only author with healthcare professional status and permission to do that, accessed the medical record of the diagnosed patients and provided this datum anonymously to the coded participant data collection. However, there was not this datum for the controls as this value is not currently measured unless there was a previous diagnostic. Given that this study lacked of any funding, to get such value by our own resources was not feasible. Nevertheless, in order to compensate for such data absence and allow to a correct participant classification, we used the fasting glucose value as it may easily be determined in current analytic controls contrary to what happens with glycated hemoglobin. We found a mean value of 5.1 ± 0.6 mmol/L for the non-diabetic participants, which falls within the range considered normal for non-diabetics (3,9–5,5 mmol/L) (IDF Diabetes Atlas, 11th Edition; Sarwar et al., 2010; Zhang et al., 2024)

Comment 4: What is the crucial impact of untargeted metabolomics approach for further routine diagnostics? Metabolomic approach is not so cheap, may be analysis of markers concentration based on classic methods would be better?

Response 4: In our opinion, the metabolomics approach has no intention of direct clinical practice for individual patients. The aim of a metabolomics approach is to find out metabolites in once that otherwise would likely need a lot of studies to be covered analytically, and with no dependence upon considerations on specific pathways and enzymes. Afterwards, a biomarker set is expected to contribute to an accurate and fast diagnosis as well as easy tracking of the disease evolution through classical analysis of the validated biomarkers. Nonetheless, metabolomics approaches can also contribute to periodically revise and improve the determined biomarker set after the metabolic pathways in which those metabolites are key players are better known. Concerning our study, it has allowed to gain insights upon verification of coincidence in validated biomarkers from previously published studies as well as to propose new biomarkers and the associated pathways, like those related to gangliosides and dipeptides.

4. Response to Comments on the Quality of English Language

Point 1: Quality of English Language

( ) The English could be improved to more clearly express the research.
(x) The English is fine and does not require any improvement.

Response 1: we thank the reviewer favorable evaluation

5. Additional clarifications

Reviewer 2 Report

Comments and Suggestions for Authors

Dear authors,

Thank you for the opportunity to review your manuscript titled "Blood metabolic biomarkers of diabetes mellitus type 2 in aged adults determined by a UPLC-MS metabolomics approach." I would like to commend you for conducting this study that aims to increase our understanding of metabolic characteristics of T2D. I believe the topic is highly relevant for other researchers and has a potential to meaningfully contribute to both the researchers community as well as to clinical practice. I have some suggestions for revisions to the text that I believe will improve the quality of the text and its impact. 

Please find my recommendations for revisions below:

General comments:

  • In the text, I recommend using more common abbreviation T2DM or T2D for type 2 diabetes mellitus, rather than DM2

Abstract:

Line 19 - I recommend rephrasing the sentence: "Type 2 Diabetes Mellitus (DM2) is a metabolic disease whose importance rises with aging." as there is no reference/source provided in the manuscript to support this statement. Prevalence of T2D certainly rises with age and it can be considered the most important risk factor of T2D, however, it can be argued that importance of T2D is also high in younger age groups as there have been a trend of increasing prevalence of T2D in younger age groups (also as a result of increasing prevalence of obesity in young)

Introduction:

  1. Line 38-39 - I recommend rephrasing the definition of T2D as the provided definition (chronic metabolic disorder with persistent hyperglycemia) is a general definition of diabetes mellitus. T2D definition should include chronicity and insulin resistance.
  2. Line 39-41 - I recommend rephrasing the sentence as this is a simplification of potential causes of increased prevalence of T2D in elderly. Various factors play a role in development of T2D in elderly, such as redistribution of fat with increased visceral adipose tissue, decrease in muscle mass (sarcopenia), decreased sensitivity of beta pancreatic cells for incretins, increased insulin resistance, reduced physical activity, accumulation of commorbidities with age (obesity, decreased kidney function, use of diabetogenic drugs)
  3. Line 44-46 - The statement regarding projection of T2D prevalence in 2045 (12.2% of population aged 20-79 yo)should apply for diabetes in general. IDF Atlas (11th Ed) would be more appropriate source for such statement.

Materials and methods

Population characteristics and recruitment

  • From the standard deviation of mean age in participatns without T2D, it seems that participants under 65 yo were also recruited - I recommend double-checking the figures whether the SD or the statement that only participants aged 65 years and over were recruited are correct.
  • I recommend adding information on duration of T2D and type of T2D medication for the participants with T2D

Discussion

  • I recommend adding a paragraph stating limitations of the study
  • I recommend adding a few sentences on potential practical implications of the findings (either for further research or clinical practice)

Author Response

For research article metabolites-3647196

Response to Reviewer 2 Comments

1. Summary

2. Questions for General Evaluation

Reviewer’s Evaluation

Response and Revisions

Yes

Can be improved

Must be improved

Not applicable

Does the introduction provide sufficient background and include all relevant references?

( )

(x)

( )

( )

Is the research design appropriate?

(x)

( )

( )

( )

Are the methods adequately described?

(x)

( )

( )

( )

Are the results clearly presented?

(x)

( )

( )

( )

Are the conclusions supported by the results?

(x)

( )

( )

( )

Are all figures and tables clear and well-presented?

( )

( )

( )

( )

Dear authors,

Thank you for the opportunity to review your manuscript titled "Blood metabolic biomarkers of diabetes mellitus type 2 in aged adults determined by a UPLC-MS metabolomics approach." I would like to commend you for conducting this study that aims to increase our understanding of metabolic characteristics of T2D. I believe the topic is highly relevant for other researchers and has a potential to meaningfully contribute to both the researchers community as well as to clinical practice. I have some suggestions for revisions to the text that I believe will improve the quality of the text and its impact. 

Please find my recommendations for revisions below:

3. Point-by-point response to Comments and Suggestions for Authors

The authors thank the reviewer’s comments which undoubtedly have contributed to improve our manuscript.

Comments 1: General comments: In the text, I recommend using more common abbreviation T2DM or T2D for type 2 diabetes mellitus, rather than DM2

Response 1: the author is right in its appointment. We have done the necessary changes.

Comments 2: Abstract:

Line 19 - I recommend rephrasing the sentence: "Type 2 Diabetes Mellitus (DM2) is a metabolic disease whose importance rises with aging." as there is no reference/source provided in the manuscript to support this statement. Prevalence of T2D certainly rises with age and it can be considered the most important risk factor of T2D, however, it can be argued that importance of T2D is also high in younger age groups as there have been a trend of increasing prevalence of T2D in younger age groups (also as a result of increasing prevalence of obesity in young)

Response 2: The authors thank this comment that fit better to a precise definition of diabetes mellitus type 2. Taking into consideration this valuable criticism together with the next one, we have combined the factors considered in both definitions to render the following definition (lines 40-44):

“ (T2DM) is a chronic metabolic disorder whose prevalence rises with age; however, its impact is also rising in younger populations owing to obesity is being increased in this population group. This disease is characterized by a marked insulin resistance and its strong deficient secretion, a situation that renders persistent insulin deficit and hyperglycaemia [1,2].” (cites 1 and 2 correspond to references IDF Diabetes Atlas, 11th edition, 2025; Lu et al., 2024)

Comment 3: Introduction:

  1. Line 38-39 - I recommend rephrasing the definition of T2D as the provided definition (chronic metabolic disorder with persistent hyperglycemia) is a general definition of diabetes mellitus. T2D definition should include chronicity and insulin resistance.
  2. Line 39-41 - I recommend rephrasing the sentence as this is a simplification of potential causes of increased prevalence of T2D in elderly. Various factors play a role in development of T2D in elderly, such as redistribution of fat with increased visceral adipose tissue, decrease in muscle mass (sarcopenia), decreased sensitivity of beta pancreatic cells for incretins, increased insulin resistance, reduced physical activity, accumulation of commorbidities with age (obesity, decreased kidney function, use of diabetogenic drugs)

3.       Line 44-46 - The statement regarding projection of T2D prevalence in 2045 (12.2% of population aged 20-79 yo)should apply for diabetes in general. IDF Atlas (11th Ed) would be more appropriate source for such statement.

Response 3: the reviewer’s suggestions to this point are very thanked and addressed accordingly as indicated below:

1.       In order to account for the reviewer’s suggestion, we have modified the text in the manuscript in the following way (lines 36-54):

“Aging of population has become a relevant concern at present in most western countries. Aging is currently associated with diverse metabolic malfunctions, which will likely lead to cardiovascular and renal diseases in the short range. Even though not exclusive to elderly, one of this age associated diseases is type 2 diabetes mellitus (T2DM), which is a chronic metabolic disorder whose prevalence rises with age; however, its impact is also rising in younger populations owing to obesity is being increased in this population group. This disease is characterized by a marked insulin resistance and its strong deficient secretion, a situation that renders persistent insulin deficit and hyperglycaemia [1,2]. a metabolic disorder characterized by persistent hyperglycemia. Elderly people tend to accumulate fat in ectopic tissues, mainly liver and skeletal muscle, a factor that may elicit insulin resistance or impaired insulin secretion and, ultimately, T2DM. Reallocation of fat that leads to an increase of the visceral adiposity is one of the multiple factors involved in the development of T2DM in elderly people, but it runs accompanied for reduction of muscular mass (sarcopenia), lower sensitivity of pancreatic beta-cells to incretins, pronounced rise of insulin resistance, decreased kidney function, lower physical activity, and co-morbidities that imply the use of drugs with a potential diabetogenic effect [1,3–5]. Thus, genetic and behavioral factors like a high carbohydrate-based nutrition together with lack of exercise predispose to T2DM development [6,7]. However, the complex molecular mechanisms involved in its development are not still well known [8].”

2.       Text modified as indicated in the previous point.

3.       The text has been modified as follows (lines 55-60): “According to data in the Diabetes Atlas published by the International Diabetes Federation (11th Edition) [1], global prevalence of diabetes in adults with age between 20 and 79 years is foreseen to reach  13% of world population in 2050 (about 852.5 million people) More than 12.2% of world population is foreseen to have T2DM in 2045, and the percentage might be higher because of not diagnosed people [9]. To reduce the economic burden of T2DM treatments is a major concern for national health systems.”

Comment 4: Materials and methods

Population characteristics and recruitment

•          From the standard deviation of mean age in participatns without T2D, it seems that participants under 65 yo were also recruited - I recommend double-checking the figures whether the SD or the statement that only participants aged 65 years and over were recruited are correct.

•          I recommend adding information on duration of T2D and type of T2D medication for the participants with T2D

Response 4: the mean age was as indicated. It seems that the majority of participants had an age close to the inferior limit of the range, a fact that led to a mean value rather low though dispersion provoked a high value of the standard deviation (SD). The values of mean, SD, median and range limit values provided directly by the software of the statistical analysis are:

Please note that the group former nomenclature is present in the image. Certainly, non-diabetic participants are somewhat older than the diabetic ones.

It is indicated in the revised version that the mean duration of the T2DM is 5 years. And medication is reported in the manuscript (2.2. Population characteristics and recruitment) by adding the following sentence:

“The medication included oral antidiabetic drugs like metformin, alone or in combination with vildagliptin, sitagliptin or dapagliflozin, or repaglinide or gliclazide, in addition to insulin.”

 Comment 5: Discussion

  • I recommend adding a paragraph stating limitations of the study
  • I recommend adding a few sentences on potential practical implications of the findings (either for further research or clinical practice)

Response 5: we consider we have conveniently addressed these valuable comments.

Regarding limitations, the paragraph that follows has been added at the end of the discussion section (lines 429-438):

“There are mostly three limitations associated to this study performance, which are: i) the cohort is rather reduced; ii) a single center recruitment though from different small villages; and iii) extensive pharmacology that is affecting both controls and patients. These limitations may have affected the biomarker search, but it is evident that a number of biomarkers found in this study are coincident with those found in other group’s studies, in particular those related to lysophospholipids. Additionally, a new class of lipids is shown in our study to afford potential biomarkers, which are the gangliosides. In spite of the burden imposed by the extensive pharmacology together with a reduced cohort, excellent group separation in the PLS-DA using two different softwares, which may account in our opinion for high robustness of our results.”

In regard to the potential implications, the following sentences have been added at the end of the Conclusion section (lines 451-454):

“The results of this study open the possibility to discriminate specific biomarkers for elderly people given that their metabolism intrinsically differs from that of the younger people and, additionally, there might be potential confounding factors bound to the polypharmacy.”

es4. Response to Comments on the Quality of English Language

Point 1: Quality of English Language

( ) The English could be improved to more clearly express the research.
(x) The English is fine and does not require any improvement.

Response 1: we thank the reviewer favorable evaluation

5. Additional clarifications

References

  1. Magliano, D.; Boyko, E.J. IDF Diabetes Atlas; 10th edition.; International Diabetes Federation: Brussels, 2021; ISBN 978-2-930229-98-0.
  2. Lu, X.; Xiel, Q.; Pan, X.; Zhang, R.; Zang, X.; Peng, G.; Zhang, Y.; Shen, S.; Tong, N. Type 2 diabetes mellitus in adults: pathogenesis, prevention and therapy. Signal Transduction and Targeted Therapy 2024, 9, 262, 10.1038/s41392-024-01951-9.
  3. Galicia-Garcia, U,; Benito-Vicente, A.; Jebari, S.; Larrea-Sebal. A.; Siddiqi, H.; Uribe, K.; Ostolaza, H.; Martín, C. Pathophysiology of Type 2 Diabetes Mellitus. Int. J. Mol. Sci. 2020, 21, 6275; doi:10.3390/ijms21176275

4. Guerrero-Fernández de Alba, I.; Orlando, V.; Monetti, V.M.; Mucherino, S.; Gimeno-Miguel, A.; Vaccaro, O.; João, M.; Poblador, B.; Prados-Torres, A.; Riccardi, G.; Menditto. Comorbidity in an Older Population with Type-2 Diabetes Mellitus: Identification of the Characteristics and Healthcare Utilization of High-Cost Patients. Frontiers in Pharmacology.2020, 11,586187; doi: 10.3389/fphar.2020.586187

Reviewer 3 Report

Comments and Suggestions for Authors

The article is devoted to a actual area - diagnostics of diseases using metabolomic data. Diabetes, studied in the presented work, is recognized as a "non-infectious epidemic" of modern humanity, so the publication will be of interest to a large number of readers. The list of references is compact and recent publications are reflected.

Some comments and questions that require clearer reflection in the text of the publication:

Lines 29-31. There are two repeated words in the sentence, one of them can be replaced, for example, with "demonstrate".

Lines 62-68. The sentence is too long with a complex structure - it would be better to rephrase it.

Lines 220-222. The figures on the upper panel and lower panel of Fig.2 could be made smaller (more compact), but without reducing the captions to the figure axes, and also align the font sizes between the figures. Then you can try to fit the figures next to each other (one on the left, the other on the right). The inscription "Scores plot" above the lower figure is not needed.

Line 262. "Gly-His correlated with MEDAS-14 (p = 0.004)" -it is not very clear in which materials this is shown (and further in this paragraph about other metabolites). Or is it not shown in the publication materials? Logically, we are talking about Table 4, but it does not have such indicators. Just as there is no mention of "consumption of sugar-rich foods". It is also not clear where the results about NDM2 are reflected. What does the line "Age" mean and its statistical values?

Line 281. In the header of table 4 there is a footnote index or something else, but it is not clear what the number 2 means: "95% CI2". The explanation for the CI decoding indicates that we are talking about 95%, which means that this value no longer needs to be indicated in the header of the table.

Line 354. Incomplete name of metabolite - "LPC(14)".

Lines 366-367. Bad sentence, the meaning is not clear. Is it like an introduction to the list of identified biomarkers?

Lines 386-391 duplicate the meaning above (382-386), need to be removed.

Author Response

For research article metabolites-3647196

Response to Reviewer 3 Comments

1. Summary

2. Questions for General Evaluation

Reviewer’s Evaluation

Response and Revisions

Yes

Can be improved

Must be improved

Not applicable

Does the introduction provide sufficient background and include all relevant references?

( )

(x)

( )

( )

Is the research design appropriate?

(x)

( )

( )

( )

Are the methods adequately described?

(x)

( )

( )

( )

Are the results clearly presented?

( )

( )

(x)

( )

Are the conclusions supported by the results?

(x)

( )

( )

( )

Are all figures and tables clear and well-presented?

( )

(x)

( )

( )

The article is devoted to a actual area - diagnostics of diseases using metabolomic data. Diabetes, studied in the presented work, is recognized as a "non-infectious epidemic" of modern humanity, so the publication will be of interest to a large number of readers. The list of references is compact and recent publications are reflected.

Some comments and questions that require clearer reflection in the text of the publication:

3. Point-by-point response to Comments and Suggestions for Authors

The authors thank the reviewer’s comments which undoubtedly have contributed to improve our manuscript.

Comments 1: Lines 29-31. There are two repeated words in the sentence, one of them can be replaced, for example, with "demonstrate".

Response 1: done

Comment 2: Lines 62-68. The sentence is too long with a complex structure - it would be better to rephrase it.

Response 2: the sentences included in this paragraph have been modified and the resulting text in the revised version, which expect to be better read, is as follows (lines 76-88):

Using untargeted first and targeted afterwards lipidomic approaches, Feng et al. [14] used two lipidomics approaches, namely an untargeted one first, and afterwards, taking into account the results of this first approach, they accomplished a targeted approach to compare the lipid profile of T2DM patients with that of healthy controls. The participants in this study were males aged 35-65 years, and the patients were classified in three groups according to their T2DM state, namely high risk, recently diagnosed and diagnosed for more than two years. The results of this study showed specific lipid species whose content is altered in the T2DM patients in comparison to the healthy controls in males aged 35-65 years according to their T2DM state, namely high risk, recently diagnosed and diagnosed for more than two years; and the lipid species they reported to have their concentration altered in T2DM individuals were mainly those bearing the phosphocholine group, ceramides, short-chain saturated triglycerides, and hydroxylinoleic cholesteryl esters.”

Comment 3: Lines 220-222. The figures on the upper panel and lower panel of Fig.2 could be made smaller (more compact), but without reducing the captions to the figure axes, and also align the font sizes between the figures. Then you can try to fit the figures next to each other (one on the left, the other on the right). The inscription "Scores plot" above the lower figure is not needed.

Response 3: We have tried to do as suggested by the reviewer, however, we can only modify the figure of the upper panel, that from the eXtended Statistics, from the original version using Adobe Illustrator. In contrast to this, we cannot modify directly the figure from the MetaboAnalyst by changing the figure settings directly in the application. Therefore, we have done the best approximations we have been able to in the PowerPoint software. The result is not exactly as expected, but we think it may now conform the reviewer’s suggestion. Nonetheless, given that the figure quality has been reduced with the modification, we have included the original separated figures in the supplementary material file.

Figure 2 as modified:

Comment 4: Line 262. "Gly-His correlated with MEDAS-14 (p = 0.004)" -it is not very clear in which materials this is shown (and further in this paragraph about other metabolites). Or is it not shown in the publication materials? Logically, we are talking about Table 4, but it does not have such indicators. Just as there is no mention of "consumption of sugar-rich foods". It is also not clear where the results about NDM2 are reflected. What does the line "Age" mean and its statistical values?

Response 4: The reviewer is right in this confounding text. We think these sentences of the text were not properly revised after a previous version was updated. We earnestly apologize for that.

Actually, the correlations of Gly-His with MEDAS-14 and of LPC(14:0) with “consumption of sugar-rich foods” were thought to be included in Table 3, not in Table 4 as indicated in the text, but a previous not-updated version of the Table 3 was mistakenly left in the submitted version of the manuscript.

In the revised version, this paragraph is split into two paragraphs, the first one (lines 303-310) still speaking of the Table 3, where the mentioned values of correlation are now included, and a second paragraph (lines 311-323) speaking of data on Table 4. The values of the statistical parameters for “consumption of sugar-rich foods” are provided as supplementary information (Supplementary Table 2). 

The line “Age” was included in the statistical analysis whose data are reported in Table 4 because we consider that when this variable was analyzed together with the metabolites could give different results to that found when it was compared alone, as it happens. Nonetheless, we now see these results are not relevant and we have decided to remove the line.

Comment 5: Line 281. In the header of table 4 there is a footnote index or something else, but it is not clear what the number 2 means: "95% CI2". The explanation for the CI decoding indicates that we are talking about 95%, which means that this value no longer needs to be indicated in the header of the table.

Response 5: we apologize for this editing mistake. In a previous versions of this table food notes were included, but in the submitted version the food notes were removed. The number 2 in CI was unproperly not deleted.  

Comment 6: Line 354. Incomplete name of metabolite - "LPC(14)".

Response 6: amended.

Comment 7: Lines 366-367. Bad sentence, the meaning is not clear. Is it like an introduction to the list of identified biomarkers?

Response 7: thanks for the advice. We have modified the sentence to read now (lines 440-442): “In this study we have determined a set of biomarkers related to type 2 diabetes mellitus (T2DM) in aged people (> 65 years) using an untargeted metabolomics study.” The intention of this sentence is to state in the onset of the Conclusion section the general aim of our study. Hope this objective is clear enough in the revised version.

Comment 8: Lines 386-391 duplicate the meaning above (382-386), need to be removed.

Response 8: this section was first fulfilled in the manuscript and in the application form afterwards, then the statements were copied from the application form to the manuscript and the previously written text was not removed. We apologize for that. Now, in the revised version, we keep the text of the application form (lines 464-469).

4. Response to Comments on the Quality of English Language

Point 1: Quality of English Language

( ) The English could be improved to more clearly express the research.

(x) The English is fine and does not require any improvement.

Response 1: we thank the reviewer favorable evaluation

5. Additional clarifications

Reviewer 4 Report

Comments and Suggestions for Authors

Metabolomics represents a powerful analytical approach that has significantly transformed our understanding of Type 2 Diabetes Mellitus. This science studies the complete set of metabolites present in biological systems, offering a comprehensive view of the biochemical processes altered in this complex metabolic disease. The application of metabolomics in diabetes research has revealed previously unknown pathophysiological mechanisms and identified potential new biomarkers for early diagnosis and patient stratification. Metabolomics also plays a crucial role in studying the interactions between diabetes and other factors such as diet, gut microbiome, and environmental factors. Metabolomic studies have demonstrated how certain nutrients or microbiota-derived metabolites can influence the development and progression of diabetes, highlighting the importance of a holistic approach in managing this disease. Though a number of research studies in this field have already been performed, novel data can produce valuable insights into diabetes mechanisms. For these reasons, I consider the manuscript worthy of publication. However, some additional discussion and corrections are required:

  1. Please add a more detailed description of analyte identification procedure, including databases used.
  2. Parameters of the mass analyzer are required.
  3. The proposed formula for unidentified compound 1 (C26H47N2O7P) is obviously erroneous. Please correct.
  4. It would be useful to perform hierarchical clustering and add a corresponding heatmap to the manuscript.

Author Response

For research article metabolites-3647196

Response to Reviewer 4 Comments

1. Summary

2. Questions for General Evaluation

Reviewer’s Evaluation

Response and Revisions

Yes

Can be improved

Must be improved

Not applicable

Does the introduction provide sufficient background and include all relevant references?

(x)

( )

( )

( )

Is the research design appropriate?

(x)

( )

( )

( )

Are the methods adequately described?

( )

( )

(x)

( )

Are the results clearly presented?

( )

(x)

( )

( )

Are the conclusions supported by the results?

(x)

( )

( )

( )

Are all figures and tables clear and well-presented?

(x)

( )

( )

( )

Metabolomics represents a powerful analytical approach that has significantly transformed our understanding of Type 2 Diabetes Mellitus. This science studies the complete set of metabolites present in biological systems, offering a comprehensive view of the biochemical processes altered in this complex metabolic disease. The application of metabolomics in diabetes research has revealed previously unknown pathophysiological mechanisms and identified potential new biomarkers for early diagnosis and patient stratification. Metabolomics also plays a crucial role in studying the interactions between diabetes and other factors such as diet, gut microbiome, and environmental factors. Metabolomic studies have demonstrated how certain nutrients or microbiota-derived metabolites can influence the development and progression of diabetes, highlighting the importance of a holistic approach in managing this disease. Though a number of research studies in this field have already been performed, novel data can produce valuable insights into diabetes mechanisms. For these reasons, I consider the manuscript worthy of publication. However, some additional discussion and corrections are required:

3. Point-by-point response to Comments and Suggestions for Authors

The authors thank the reviewer’s comments which undoubtedly have contributed to improve our manuscript.

Comments 1: Please add a more detailed description of analyte identification procedure, including databases used.

Response 1: the following sentences have been added now in the 2.3.3. UPLC-MS data analysis section (lines 172-187):

“Values of m/z with a defined chromatographic peak were only accepted to potentially arise from any true compound. Selected differential metabolites were tentatively identified by comparison of the elemental composition measured in the mass spectrum with that available in the free accessible databases HMDB, METLIN, KEGG, ChEBI and LipidMaps. Additionally, comparison of relative retention time and adduct formation for those compounds that had been identified in previous studies (lysophosphatidyl cholines for instance) was taken into consideration. Specific fragments in the MSE spectrum, either known (m/z 184.07 and the fragment corresponding to the neutral loss of the acyl group for phosphatidyl cholines and sphingomyelins, for instance) or visualized in the HMDB for the given compound were used to validate the compound identification. For gangliosides, the nominal mass (M) calculated from the m/z value of the spectrum first peak and taking into consideration that ionization was +2 (i.e. [M+X+Y]2+, where X and Y may be equal or different and account for H, NH4+ or both) was used to find out whether it corresponded to any compound in the LipidMaps database, but a specific species could not be determined and only some are tentatively suggested as indicated where appropriate.”

We expect this added explanation will conform the reviewer’s request.

Comment 2: Parameters of the mass analyzer are required.

Response 2: some parameters of the mass analyzer were already provided in the former version (lines 152-155):

“The metabolites were detected with positive ionization, and the parameters were capillarity 2.5 KV, cone 25 V, source temperature 100 ºC, desolvation temperature 320 ºC, cone gas (nitrogen) 35 L/h, and desolvation gas flow (nitrogen) 700 L/h.”

In the revised version, the following sentences have been added (lines 144-152):

“A MSE method was used for the analysis, in which data were acquired within the m/z range of 100 to 1200 Da under the low energy function (function 1), it being full-scan equivalent, and within the m/z range of 50 to 1000 Da under the high energy function (function 2) with non-specific fragmentation of base peak m/z values detected in the full-scan. Data were acquired in resolution mode (expected error of less than 3 ppm corresponding to a minimal resolution of 20,000) using the MassLynx® software (WATERS, Manchester, UK). The QToF-MS was calibrated using 0.5 mM sodium formate in 9:1 (v/v) 2-propanol:water, and as reference 2 ng/µL Leucine-Enkephalin (Leu-Enk) in 50:50 (v/v) acetonitrile:water with 0.1% formic acid was used.”

Comment 3: The proposed formula for unidentified compound 1 (C26H47N2O7P) is obviously erroneous. Please correct.

Response 3: the reviewer is right. Thanks for the warning.

Certainly, the formula shown in the table is erroneous, it corresponded to another compound that was finally rejected, and actually we cannot propose a formula for this compound because it cannot be obtained with the MassLynx software due to the double ionization (+2). Therefore, we have indicated in the table caption the following:

“Gangliosides and unidentified 1 have ionization +2 (]M+2H]2+, [M+H+NH4]2+, or [M+2NH4]2+). A formula for unidentified 1 cannot be proposed.”

Using the same method as for gangliosides, we obtained only compounds that are not expected to be present in human plasma, namely Ginsenosides (C53H90O22) or Glycerophosphoinositolglycans (C51H95O18P)

Comment 4: It would be useful to perform hierarchical clustering and add a corresponding heatmap to the manuscript.

Response 4: we have addressed this suggestion and the results have been incorporated in the revised manuscript. A figure (3) has been added to show the clustering tree and the corresponding heatmap. A few lines of text have also been added to comment the results of this analysis as well as to discuss them. See below:

IN RESULTS (lines 330-334): “Hierarchical cluster analysis was also conducted for the selected metabolites illustrated in Table 2 (Figure 3). The general cluster was split into two major sub-clusters, which included small clusters containing samples from both T2DM and NT2DM2 groups. This partial mixing may be a reflection of the data dispersion found for some metabolites, a fact that leads to partial intersection of both groups. The partial intersection is also reflected in the heatmap, but enhanced metabolite content is shown to happen in the NT2DM group.”

IN DISCUSSION (lines 423-428):_” Even though data dispersion, a feature common to plasma analysis, may have led to partial mixing of samples in the hierarchical cluster analysis (Figure 3), the majority of small sub-cluster showed grouping of samples within the same NT2DM or T2DM groups, with apparent enhanced content of the metabolites in the NT2DM group. It should be taken into consideration that not all the metabolites rendered statistically significant differences.”

Figure 3. Clustering tree for the samples according to the metabolite contents (left panel). The corresponding heatmap is displayed in the right panel.

4. Response to Comments on the Quality of English Language

Point 1: Quality of English Language

( ) The English could be improved to more clearly express the research.

(x) The English is fine and does not require any improvement.

Response 1: we thank the reviewer favorable evaluation

5. Additional clarifications

Round 2

Reviewer 1 Report

Comments and Suggestions for Authors

Ok, manuscript can be accepted for publication

Reviewer 4 Report

Comments and Suggestions for Authors

The authors have addressed all issues and the manuscript can be published in my opinion